# Differences in the Perceived Likelihood of Receiving COVID-19 Vaccine

**DOI:** 10.3390/ijerph192113723

**Published:** 2022-10-22

**Authors:** David Adzrago, Saanie Sulley, Cameron K. Ormiston, Lohuwa Mamudu, Faustine Williams

**Affiliations:** 1Center for Health Promotion and Prevention Research, The University of Texas School of Public Health, The University of Texas Health Science Center at Houston, Houston, TX 77030, USA; 2National Healthy Start Association, Washington, DC 20005, USA; 3Division of Intramural Research, National Institute on Minority Health and Health Disparities, National Institutes of Health, Bethesda, MD 20892, USA; 4Department of Public Health, California State University, Fullerton, CA 92831, USA

**Keywords:** COVID-19 pandemic, COVID-19 vaccine, prevention, public health, vaccine hesitancy

## Abstract

There are limited studies on the perceived likelihood of receiving a COVID-19 vaccine among the general US population and its subpopulations. We examined the association between the perceived likelihood of receiving a COVID-19 vaccine with the self-reported likelihood of contracting COVID-19, social-distancing stress, COVID-19 diagnosis status, mental health disorders, and sociodemographic characteristics. The data were collected using a national cross-sectional survey (N = 5404) between 13 May 2021 and 9 January 2022. A multivariable logistic regression analysis was performed. Setting: United States. Participants: Adults aged ≥ 18 years. The majority of US adults (67.34%) indicated they intended to receive a COVID-19 vaccine. There was a decreased perceived likelihood of getting vaccinated associated with those aged 18–49 years (Adjusted Odds Ratio (AOR) = 0.29–59; 95% Confidence Interval (CI) = 0.20–0.85); with a less than college education (AOR = 0.37–58; 95% CI = 0.28–0.68); with no health insurance (AOR = 0.48; 95% CI = 0.40, 0.58); with no perceived likelihood of contracting COVID-19 (AOR = 0.78; 95% CI = 0.68, 0.89); and with anxiety/depression (AOR = 0.67; 95% CI = 0.59, 0.76). Black/African Americans had a lower perceived likelihood of receiving a COVID-19 vaccine (AOR = 0.84; 95% CI = 0.71, 0.98), while Asians (AOR = 1.92; 95% CI = 1.35, 2.74) and Hispanics/Latinos (AOR = 1.34; 95% CI = 1.03, 1.74) had a higher perceived likelihood compared with Whites. Individuals reporting social distancing as stressful (AOR = 1.21; 95% CI = 1.01, 1.45) were associated with an increased perceive likelihood of receiving a COVID-19 vaccine. Our study showed that younger adults, Black/African Americans, and those with a less than college education, no health insurance, or anxiety/depression may be less likely to receive vaccination. Future research should examine the explanatory mechanisms contributing to the lower perceived likelihood of vaccination among these groups, such as barriers to vaccine education or vaccine access. Public health interventions should prioritize these populations to improve vaccination rates.

## 1. Introduction

MacDonald et al. [1] define vaccine hesitancy as a “delay in acceptance or refusal of vaccines despite availability of vaccination services”. Vaccine hesitancy can lead to a resurgence of preventable diseases, causing avoidable public health expenditures and disease morbidity [2]. The interest in addressing vaccine hesitancy has increased recently due to the COVID-19 pandemic and difficulties in containing the spread of the virus [3,4]. Skepticism and mistrust toward the COVID-19 vaccines exist for several reasons, including the pace of vaccine development, political party affiliation, disinformation, possible side effects, and its effectiveness [3,4,5,6,7,8]. Although COVID-19 vaccines are free of cost and widely available in the United States (US), around 25% of the US population had not received at least one dose of the COVID-19 vaccine as of July 2022, indicating an urgent need to further our understanding of COVID-19 vaccine hesitancy among the US population and better inform public policy and public health interventions [7]. Studies exploring vaccine willingness among specific populations, such as younger adults and women, suggest attitudes towards vaccines, the perceived risk of infection, and the perceived risk of vaccines are predictive of vaccine hesitancy [6,9,10,11,12,13]. Furthermore, research from Williams et al. [12] has identified disparities in COVID-19 vaccination coverage and access among US adults, with Black, Hispanic/Latino, and financially insecure individuals being particularly at risk for low vaccination rates. Regional differences in vaccine rates have also been reported, with rural areas having lower vaccine rates compared to urban areas [14]. Understanding the diverse factors that impact the likelihood of vaccination is imperative in preparing public health officials, health providers, and the government to prepare and tailor educational programs for current and future infectious disease emergencies.

Additionally, vaccine hesitancy has been linked with an elevated risk of depression and anxiety, and greater stress is negatively associated with vaccination willingness/likelihood [15,16]. Getting the COVID-19 vaccine, however, may reduce depression and anxiety symptoms, particularly among the most vulnerable, such as individuals who are categorized as low-income, non-teleworking, or who have children, making the improvement of vaccination rates important to both physical and mental health [17,18]. Given that our understanding of the association between mental health conditions and vaccine hesitancy is limited, further investigation of the perceived COVID-19 vaccination likelihood among those with depression/anxiety is, therefore, an important public health issue.

Our study examined the association between the perceived likelihood of receiving a COVID-19 vaccine and COVID-19 diagnosis status, the perceived likelihood of contracting COVID-19, mental health disorder symptoms, and sociodemographic characteristics. Improving our understanding of these associations can aid in identifying priority populations and developing effective community-based strategies to improve vaccine education.

## 2. Methods

### 2.1. Study Design and Samples

Our study utilizes data from a larger cross-sectional survey that assessed general health status; COVID-19 symptoms, testing, and prevention; chronic illness management; mental health; the pandemic’s economic impact; and participants’ sociodemographic characteristics. It was an online survey conducted among a national sample of US adults aged 18 years or older. This survey sought to (1) examine the effects of social and physical distancing on physical and psychosocial health and disparities among US adults (i.e., US-born and foreign-born), and (2) identify factors that potentially influence/explain the disparities.

The National Institutes of Health contracted Qualtrics LLC to conduct the recruitment and distribution of the online survey to the participants. Qualtrics used proprietary consumer panels to randomly sample participants with matching demographic characteristics to complete the survey. Oversampling was applied for low-income (<USD 25,000 annual household income) and rural (self-reported to reside in rural area, cross-referenced with zip codes already collected by Qualtrics) adults among non-Hispanic White, non-Hispanic Black, Hispanic, and foreign-born participants. The survey was only available in English. A total of 10,000 surveys were distributed between 13 May 2021 and 9 January 2022, and the research team received 5938 surveys from Qualtrics. Information Management Services, Inc. (IMS), a research support firm that provides analytic services, was given the de-identified survey data to clean and manage. To improve study’s integrity, initial data cleaning by IMS included flagging surveys based on completion rate and survey completion time. Participants were flagged and removed from analysis if they completed less than 80% of the survey based on 102 questions after accounting for skip pattern items and those that took less than 5 min to complete the survey. The study ended with a total of 5413 samples. Our current analysis included 5404 samples with complete data on response to the perceived likelihood of getting vaccinated with the COVID-19 vaccine survey question.

### 2.2. Measures

#### 2.2.1. Primary Outcome

The dependent variable/outcome of this study was the perceived likelihood to receive a COVID-19 vaccine. Respondents were asked, “Now that we have a Coronavirus/COVID-19 vaccine available, have you been/planning to be vaccinated?” The response options included not at all likely, slightly likely, moderately likely, very likely, and extremely likely. We dichotomized this variable into not at all likely and likely (i.e., slightly likely, moderately likely, very likely, and extremely likely).

#### 2.2.2. Explanatory/Independent Variables

The main independent variables included self-reported likelihood of contracting COVID-19, COVID-19 diagnosis status, anxiety or depression symptoms, and post-traumatic stress disorder (PTSD) symptoms. We assessed self-reported likelihood of contracting COVID-19 by asking the question, “Given your overall self-rated health, how likely do you feel that you will contract Coronavirus/COVID-19?”. The response options were not at all likely, slightly likely, moderately likely, very likely, and extremely likely. This variable was also converted into a binary variable, as performed for the outcome variable. To determine COVID-19 diagnosis status, participants were asked, “Have you been tested for Coronavirus/COVID-19?” (Yes/No). If the answer was yes, they were asked a follow-up question on the result of that test (“Was the test for Coronavirus/COVID-19 positive?” (Yes/No/Don’t know)).

Anxiety and depression variables were derived from the questions that measured anxiety using the Generalized Anxiety Disorder 2-item (GAD-2) and depression using Patient Health Questionnaire-2 (PHQ-2) [19,20]. The total score for GAD-2 ranges from 0–6, and a score of ≥3 indicates anxiety; similar scores are determined for depression on the PHQ-2 scale. The GAD-2 and PHQ-2 were combined to form the PHQ-4. The pooled total scores on GAD-2 and PHQ-2 range from 0–12, and scores of ≥3 suggest anxiety/depression. Hence, we analyzed our study’s levels of anxiety/depression based on PHQ-4. PTSD symptoms were assessed by asking the participants, “Sometimes things happen to people that are unusually or especially frightening, horrible, or traumatic. For example, a serious accident or fire, a physical or sexual assault or abuse, a war, etc. Have you ever experienced this kind of event?” (Yes/No) [21].

The covariates included sociodemographic characteristics such as age, gender identity, sexual orientation, race/ethnicity, level of education completed, marital status, annual household income, health insurance status, and US census region. Sexual orientation was included as a variable given the historical medical mistreatment and abuse faced by sexual minority groups that have been shown to affect COVID-19 vaccine uptake [22]. General physical health status was also included. Additionally, household members’ COVID-19 diagnosis status and social-distancing stress were included in the study. The household members’ COVID-19 diagnosis status was assessed by asking, “Has anyone in your household or residence tested positive for Coronavirus/COVID-19?” (Yes/No). Social-distancing stress was assessed by asking, “How stressful has socially distancing been for you?”, with response options including very stressful, somewhat stressful, a little stressful, and not at all stressful. We recoded this variable into a binary as stressful (very stressful, somewhat stressful, and a little stressful) and not at all stressful.

### 2.3. Statistical Analysis

We conducted descriptive and bivariate analyses to estimate the prevalence and statistical differences in the perceived likelihood of getting vaccinated with a COVID-19 vaccine according to sociodemographic characteristics, general physical health status, the likelihood of contracting COVID-19, COVID-19 status, social-distancing stress, anxiety/depression, and PTSD. We also conducted multivariable logistic regression analyses to assess the associations represented by three models. Chi-squared tests were used for the bivariate analysis to test group differences. In Model I, we assessed the association between the perceived likelihood of getting vaccinated with a COVID-19 vaccine and sociodemographic characteristics. Model II assessed the association between the perceived likelihood of getting vaccinated with a COVID-19 vaccine and general physical health status, the likelihood of contracting COVID-19, COVID-19 status, social-distancing stress, and anxiety/depression. In Model III, we combined Models I and II to assess the perceived likelihood of getting vaccinated with a COVID-19 vaccine. The models were based on complete case analyses. All the analyses were conducted using STATA/SE 16.1. The results are reported using frequencies, percentages, adjusted odds ratios (AOR), 95% confidence interval (CI) at the 2-tailed level, and a statistical significance level of *p* < 0.05.

## 3. Results

Of the 5404 participants, the highest proportions were those aged 35–49 years (21.93%), women (62.55%), heterosexuals (89.28%), White (42.56%), those who had a college degree or higher (38.72%), individuals that were married/living with a partner (54.34%), those who had an annual household income of ≥USD 75,000 (26.26%), individuals with health insurance (86.52%), and individuals that resided in the Southern region of the US (44.62%) (Table 1). Among the participants in the study sample, 67.34% reported they were likely to receive a COVID-19 vaccine (see Table 1).

Bivariate analysis showed that all the variables, except the COVID-19 diagnosis status of the survey participant and household member(s), were significantly associated with the perceived likelihood of getting vaccinated with a COVID-19 vaccine. In addition, 32.78% of those aged 26–34 years, 37.37% of lesbian and 35.92% of bisexual individuals, 41.87% of Black/African Americans, 53.12% of those with <high school education, 43.09% of those with <USD 25,000 annual household income, and 36.11% of those with a fair/poor general health status reported they were not likely to receive a COVID-19 vaccine (see Table 1).

In Model I, individuals aged 18–49 years had 40–70% lower odds of reporting COVID-19 vaccination intention compared with those aged ≥ 50 years. Model III showed comparable results. In Models I (AOR = 1.90; 95% CI = 1.11, 3.28) and III (AOR = 2.05; 95% CI = 1.18, 3.57), gay individuals were more likely to report COVID-19 vaccination intention compared to their heterosexual counterparts. Black/African Americans, compared to Whites, had a lower perceived likelihood of receiving a COVID-19 vaccine (Model I: AOR = 0.82; 95% CI = 0.70, 0.96 and Model III: AOR = 0.84; 95% CI = 0.71, 0.98). Asians (Model I: AOR = 2.05; 95% CI = 1.58, 2.64 and Model III: AOR = 2.14; 95% CI = 1.66, 2.78) and Latinos/Hispanics (Model I: AOR = 1.39; 95% CI = 1.16, 1.67 and Model III: AOR = 1.42; 95% CI = 1.18, 1.71) were more likely to report an intention to receive a COVID-19 vaccine compared with Whites.

In both Models I and III, the odds of expressing a COVID-19 vaccination intent were 42–64% (AORs ranged from 0.36 to 0.58) lower for those with less than a college education compared with those with a college or higher degree. Individuals with annual household income of <USD 50,000 were 19–44% less likely to report COVID-19 vaccination intent compared with those with ≥USD 75,000 annual household income in both Models I and III. Those with no health insurance had 52% (AOR = 0.48; 95% CI = 0.48, 0.58) lower odds of having an intention to receive a COVID-19 vaccine compared with those with health insurance.

Model II showed that individuals who reported not being likely to contract COVID-19 were 29% (AOR = 0.71; 95% CI = 0.63, 0.80) less likely to express COVID-19 vaccination intent compared with those likely to contract COVID-19. Living in a household with an individual who tested positive for COVID-19 was associated with a 16% (AOR = 0.84; 95% CI, 0.71, 0.98) lower perceived likelihood of getting vaccinated than those not living in a household with an infected individual. Those who reported social distancing as stressful were 16–21% more likely to express an intention to get vaccinated compared to those who reported social distancing as not stressful in Models II and III. Individuals experiencing anxiety/depression symptoms, compared with those not experiencing these symptoms, were 33% less likely to report an intention to be vaccinated (AOR = 0.67; 95% CI = 0.59, 0.76) in Model II. See Table 2 for further details.

## 4. Discussion

This study examined the association between the perceived likelihood/intention of receiving a COVID-19 vaccine and the likelihood of contracting COVID-19, sociodemographic factors, and mental health symptoms. Consistent with the findings of other studies, we found that the majority (67.34%) of participants were likely to get vaccinated [23,24]. In addition, most of the participants found social distancing to be stressful (68.80%), which further highlights the potential adverse effects of social distancing on mental health and stress [25,26,27]. In general, the perceived likelihood of receiving a COVID-19 vaccine was associated with age, sexual orientation, race/ethnicity, the level of education, income, the likelihood of contracting COVID-19, and anxiety/depression.

Around 31–33% of those who perceived that they were unlikely to be vaccinated were young (aged 18–34 years), which is a potential area of concern given that this age group is less likely to perform COVID-19-mitigating behaviors [28,29]. The lower perceived likelihood of getting vaccinated among younger adults may be due to a perceived lower risk of infection and severe symptoms [28]. We also found that gay respondents were more likely to report vaccination intent than heterosexual individuals. However, it should be noted that their high perceived likelihood of receiving the vaccine may not be synonymous with vaccine access or eventual vaccination given that sexual minority communities experience greater health barriers than heterosexual individuals [30,31,32]. Specifically, sexual minority individuals experience higher rates of stigmatization, medical abuse, harassment, and medical mistrust due to historical medical mistreatment, health services or information often not being tailored to their needs, and the lack of safe, affirming clinical and public spaces—all of which have been linked with vaccine hesitancy among sexual minority groups [22,31,32]. Further studies, particularly longitudinal studies, with larger samples of sexual minority individuals should examine these issues further.

We found that Black/African American participants were 16–18% less likely to report vaccination intention, which is consistent with prior studies [4,33,34]. The well-documented Black/African American hesitancy and wariness towards healthcare professionals, institutions, and experimental drugs is due to historical discrimination, exploitation, and abuse by the medical and healthcare industry and community [33,35]. Diverse approaches, including concerted educational initiatives, which address these justified concerns are, therefore, vital to improving vaccine uptake rates [4]. Consistent with the literature [12,36,37,38], we found that lower educational status and income individuals had a lower perceived likelihood of getting vaccinated for COVID-19. In fact, as of December 2021, more than half of unvaccinated individuals in US households were low-income and had lower education levels [39]. This further highlights the need to re-evaluate vaccination strategies to promote trust in vaccines in addition to innovative ideas such as mobile vaccination clinics and other incentives to ensure equitable access to vaccines regardless of income, educational status, or location.

Additionally, we found that Asians and Hispanics/Latinos were more likely to perceive themselves as getting vaccinated compared to Whites. These findings are consistent with the existing literature; however, these studies utilized a specific patient population [7,40]. For example, a study utilizing a Veteran Health Administration cohort found that Hispanic/Latino and Asian individuals were more likely to perceive themselves receiving the vaccine compared to Whites. We further demonstrate this trend in a national sample. Though our results indicate an increased perceived likelihood to receive a COVID-19 vaccine among Hispanics/Latinos, data from April 2021 show that vaccination rates are significantly lower for Hispanics/Latinos compared to Whites [41]. This suggests Hispanics/Latinos face significant barriers to obtaining the vaccine despite their increased willingness and underscores the urgent need to improve vaccine equity [41]. Though the rates of receiving at least one dose among Hispanics/Latinos are increasing over time and are now higher than Whites—a trend that may be attributed to public health outreach and educational approaches—disparities persist in the rates of receiving booster shots [7,42,43,44].

Participants reporting a lower perceived likelihood of contracting COVID-19 had significantly lower odds of endorsing vaccination willingness. This may be because of a lower perceived need for immunization due to a lower perceived risk of infection. Indeed, lower risk perception is associated with lower protective behaviors [45]. Although Patterson et al. [10] found no association between COVID-19 vaccine willingness and perceived COVID-19 risk, it should be noted that this result was among a sample of rural central Kentuckians, and vaccine willingness was low regardless of the perceived risk. Thus, Patterson et al.’s [10] finding may not be generalizable to the US population. Our study, however, utilizes a national sample. We also found the perceived likelihood of vaccination to be higher among individuals who found social distancing to be stressful. Prior research suggests feelings of loneliness or stress due to social distancing are a barrier to adhering to stay-at-home guidelines [46]. Therefore, individuals reporting social distancing-related stress may see the COVID-19 vaccine as a gateway to socializing with others and avoiding social-distancing protocols. Indeed, Chen et al. [25] posited that COVID-19 vaccination may reduce psychological distress resulting from quarantining.

It is also interesting to note that about 39% of individuals who perceived themselves as unlikely to get vaccinated had anxiety/depression, providing insight into the potential impacts of mental health on COVID-19 vaccine uptake [16]. Thus, consistent with Perlis et al. [47], we found that individuals experiencing depression and anxiety were less likely to perceive themselves receiving a vaccine, highlighting the need for tailored outreach approaches to meet both vaccination needs and mental health services for this population. Although we are unable to define the mechanism behind this association, depression/anxiety has been linked to a higher susceptibility to COVID-19 misinformation and higher mistrust of health institutions regarding COVID-19 [47]. Furthermore, given the increase in depression, anxiety, psychological distress, and other mental symptoms due to the pandemic, interventions addressing mental health are necessary [25,48,49].

Notwithstanding these findings, there are some limitations to consider. The data were from a cross-sectional survey that provided rich information on the perceived likelihood of COVID-19 vaccination, depression/anxiety, diagnosis status, and other COVID-19-related outcomes among a large US sample. The sample, however, was not representative of the US. In addition, we cannot determine causality or temporal directionality due to the cross-sectional nature of the sample. The study survey was also only available in English and online, meaning individuals with limited English proficiency and individuals without access to the Internet might not have been represented. Also, the survey was conducted in the general population regardless of their COVID-19 vaccination status. As such, individuals who had already received the vaccine could also have been represented. Furthermore, race/ethnicity was defined using aggregated groups, making the exact generalization of our analyses on racial/ethnic groups difficult. Thus, future studies should look to analyze the perceived likelihood of receiving a COVID-19 vaccine and vaccination rates among Hispanic/Latino and Asian heritage groups. For example, initial research suggests differences in vaccination coverage by Asian subgroups [50]. The disaggregation of data can help ensure effectiveness of culturally appropriate public health interventions and vaccination awareness and promotion campaigns. Finally, there were large missing age cases and we analyzed them as a category of because we could not ascertain the reason behind their absence. These missing cases of age could have over- or underestimated the results for the other age categories.

## 5. Conclusions

The present study found differences in the perceived likelihood to receive a COVID-19 vaccine across a range of different factors, including age, education, income, insurance status, race/ethnicity, social-distancing stress, and perceived risk of SARS-CoV-2 infection. Our findings contribute to the growing body of literature on COVID-19 vaccine hesitancy by highlighting populations that may be more likely to forgo receiving a vaccine, such as younger adults, individuals with lower income and education, and those with a lower perceived risk of infection. Barriers to vaccine access must be addressed as well, given that Hispanic/Latino respondents exhibited a significantly higher perceived likelihood of receiving the vaccine, yet their vaccination rates lagged behind White individuals for much of the initial vaccine rollout and continue to do so with respect to booster shots [7]. Ultimately, our findings can inform COVID-19 vaccine promotion programs with respect to targeting specific populations who may be less likely to receive a COVID-19 vaccine as we strive toward a post-pandemic world.

## Figures and Tables

**Table 1 ijerph-19-13723-t001:** Descriptive and bivariate analyses of the likelihood of getting vaccinated according to sociodemographic status, likelihood of contracting COVID-19, and mental health disorder symptoms (N = 5404).

	Overall Sample	Not at All Likely	Likely	*p*-Value
	**N (%)**	**n (%)**	**n (%)**	
		1765 (32.66)	3639 (67.34)	
**Age groups**				<0.001
18–25	1101 (20.37)	339 (30.79)	762 (69.21)	
26–34	729 (13.49)	239 (32.78)	490 (67.22)	
35–49	1185 (21.93)	265 (22.36)	920 (77.64)	
≥50	356 (6.59)	47 (13.20)	309 (86.80)	
Missing	2033 (37.62)	875 (43.04)	1158 (56.96)	
**Gender identity**				0.047
Man	1904 (35.28)	572 (30.04)	1332 (69.96)	
Non-Binary	56 (1.04)	19 (33.93)	37 (66.07)	
Something else	18 (0.33)	8 (44.44)	10 (55.56)	
Transgender	43 (0.80)	15 (34.88)	28 (65.12)	
Woman	3376 (62.55)	1145 (33.92)	2231 (66.08)	
**Sexual orientation**				0.007
Bisexual	309 (5.75)	111 (35.92)	198 (64.08)	
Gay	97 (1.81)	19 (19.59)	78 (80.41)	
Heterosexual	4797 (89.28)	1556 (32.44)	3241 (67.56)	
Lesbian	99 (1.84)	37 (37.37)	62 (62.63)	
Something else	71 (1.32)	31 (43.66)	40 (56.34)	
**Race/ethnicity**				<0.001
Asian	561 (10.38)	101 (18.00)	460 (82.00)	
Black/African American	1340 (24.80)	561 (41.87)	779 (58.13)	
Latino/Hispanic	982 (18.17)	311 (31.67)	671 (68.33)	
Something else	221 (4.09)	88 (39.82)	133 (60.18)	
White	2300 (42.56)	704 (30.61)	1596 (69.39)	
**Level of education completed**				<0.001
Less than High School	320 (5.93)	170 (53.12)	150 (46.88)	
High school diploma or GED	1244 (23.06)	542 (43.57)	702 (56.43)	
Some college/vocational or technical school	1742 (32.29)	624 (35.82)	1118 (64.18)	
College or higher degree	2089 (38.72)	426 (20.39)	1663 (79.61)	
**Marital status**				<0.001
Divorced	533 (9.90)	152 (28.52)	381 (71.48)	
Married/Living with a partner	2927 (54.34)	878 (30.00)	2049 (70.00)	
Never been married	1639 (30.43)	643 (39.23)	996 (60.77)	
Separated	113 (2.10)	36 (31.86)	77 (68.14)	
Widowed	174 (3.23)	53 (30.46)	121 (69.54)	
**Annual household income**				<0.001
<USD 25,000	1302 (24.35)	561 (43.09)	741 (56.91)	
USD 25,000 to <USD 35,000	819 (15.32)	324 (39.56)	495 (60.44)	
USD 35,000 to <USD 50,000	829 (15.50)	307 (37.03)	522 (62.97)	
USD 50,000 to <USD 75,000	993 (18.57)	280 (28.20)	713 (71.80)	
≥USD 75,000	1404 (26.26)	274 (19.52)	1130 (80.48)	
**Health insurance status**				<0.001
No	728 (13.48)	394 (54.12)	334 (45.88)	
Yes	4671 (86.52)	1369 (29.31)	3302 (70.69)	
**US census region**				<0.001
Midwest	921 (17.19)	315 (34.20)	606 (65.80)	
Northeast	852 (15.90)	249 (29.23)	603 (70.77)	
South	2391 (44.62)	880 (36.80)	1511 (63.20)	
West	1195 (22.30)	302 (25.27)	893 (74.73)	
**General physical health status**				0.004
Excellent/good	4257 (78.86)	1348 (31.67)	2909 (68.33)	
Fair/poor	1141 (21.14)	412 (36.11)	729 (63.89)	
**Perceived likelihood of contracting COVID-19**				<0.001
Not at all likely	1702 (31.55)	639 (37.54)	1063 (62.46)	
Likely	3693 (68.45)	1122 (30.38)	2571 (69.62)	
**COVID-19 diagnosis status**				0.598
No	4901 (90.69)	1606 (32.77)	3295 (67.23)	
Yes	503 (9.31)	159 (31.61)	344 (68.39)	
**Household member positive COVID-19 diagnosis status**				0.077
No	4374 (81.11)	1406 (32.14)	2968 (67.86)	
Yes	1019 (18.89)	357 (35.03)	662 (64.97)	
**Social-distancing stress**				0.017
Not at all stressful	1684 (31.20)	588 (34.92)	1096 (65.08)	
Stressful	3713 (68.80)	1174 (31.62)	2539 (68.38)	
**Depression symptoms (PHQ-2)**				<0.001
No	3952 (73.44)	1208 (30.57)	2744 (69.43)	
Yes	1429 (26.56)	549 (38.42)	880 (61.58)	
**Anxiety symptoms (GAD-2)**				<0.001
No	3905 (73.06)	1165 (29.83)	2740 (70.17)	
Yes	1440 (26.94)	573 (39.79)	867 (60.21)	
**Anxiety/depression**				<0.001
No	3631 (67.22)	1085 (29.88)	2546 (70.12)	
Yes	1771 (32.78)	679 (38.34)	1092 (61.66)	
**PTSD-5 qualifying status**				0.002
No	3707 (68.61)	1162 (31.35)	2545 (68.65)	
Yes	1696 (31.39)	603 (35.55)	1093 (64.45)	

GAD-2 = Generalized Anxiety Disorder. PHQ-2 = Patient Health Questionnaire-2. PHQ-4 = Patient Health Questionnaire-4. PTSD = Posttraumatic Stress Disorder. All *p*-values are based on chi-square tests for the categorical variables. Statistical significance at *p* < 0.05. Each category does not add up to overall total due to missing data.

**Table 2 ijerph-19-13723-t002:** Multivariable logistic regression analysis of the perceived likelihood of getting vaccinated and its associated sociodemographic, likelihood of contracting COVID-19, and mental health disorder factors.

	Model I (n = 5245)	Model II (n = 5371)	Model III (n = 5214)
	AOR (95% CI)	AOR (95% CI)	AOR (95% CI)
**Age groups**			
18–25	0.30 *** (0.21, 0.44)		0.29 *** (0.20, 0.42)
26–34	0.32 *** (0.22, 0.47)		0.32 *** (0.22, 0.46)
35–49	0.60 ** (0.42, 0.86)		0.59 ** (0.41, 0.85)
≥50	Ref		
Missing	0.23 *** (0.16, 0.32)		0.22 *** (0.16, 0.32)
**Gender identity**			
Man	1.04 (0.91, 1.19)		1.05 (0.91, 1.20)
Non-Binary	1.46 (0.75, 2.84)		1.52 (0.78, 2.96)
Something else	1.20 (0.40, 3.65)		1.31 (0.43, 4.01)
Transgender	1.17 (0.57, 2.39)		1.38 (0.66, 2.88)
Woman	Ref		-
**Sexual orientation**			
Bisexual	1.24 (0.95, 1.62)		1.24 (0.95, 1.63)
Gay	1.90 * (1.11, 3.28)		2.05 * (1.18, 3.57)
Heterosexual	Ref		-
Lesbian	1.17 (0.75, 1.85)		1.21 (0.77, 1.91)
Something else	1.00 (0.56, 1.78)		0.97 (0.54, 1.74)
**Race/ethnicity**			
Asian	2.05 *** (1.58, 2.64)		2.14 *** (1.66, 2.78)
Black/African American	0.82 * (0.70, 0.96)		0.84 * (0.71, 0.98)
Latino/Hispanic	1.39 *** (1.16, 1.67)		1.42 *** (1.18, 1.71)
Something else	0.78 (0.57, 1.06)		0.80 (0.58, 1.10)
White	Ref		-
**Level of education completed**			
Less than High School	0.36 *** (0.28, 0.48)		0.37 *** (0.28, 0.49)
High school diploma or GED	0.49 *** (0.41, 0.59)		0.50 *** (0.42, 0.60)
Some college/vocational or technical school	0.57 *** (0.49, 0.67)		0.58 *** (0.49, 0.68)
College or higher degree	Ref		-
**Marital status**			
Divorced	1.18 (0.94, 1.48)		1.21 (0.96, 1.52)
Married/Living with a partner	Ref		-
Never been married	1.20 * (1.03, 1.40)		1.24 ** (1.06, 1.45)
Separated	1.61 * (1.04, 2.49)		1.67 * (1.07, 2.59)
Widowed	0.96 (0.66, 1.41)		1.03 (0.70, 1.51)
**Annual household income**			
<USD 25,000	0.56 *** (0.45, 0.69)		0.57 *** (0.45, 0.70)
USD 25,000 to <USD 35,000	0.57 *** (0.45, 0.71)		0.56 *** (0.45, 0.71)
USD 35,000 to <USD 50,000	0.59 *** (0.47, 0.73)		0.57 *** (0.46, 0.71)
USD 50,000 to <USD 75,000	0.81 (0.66, 1.00)		0.81 * (0.66, 0.99)
≥USD 75,000	Ref		-
**Health insurance status**			
No	0.48 *** (0.40, 0.57)		0.48 *** (0.40, 0.58)
Yes	Ref		-
**US census region**			
Midwest	0.83 (0.67, 1.03)		0.48 (0.67, 1.04)
Northeast	Ref		-
South	0.82 * (0.68, 0.99)		0.84 (0.69, 1.01)
West	1.09 (0.88, 1.35)		1.10 (0.88, 1.36)
**General physical health status**			
Excellent/good		1.13 (0.98, 1.31)	0.97 (0.83, 1.13)
Fair/poor	Ref	-	-
**Perceived likelihood of contracting COVID-19**			
Not at all likely		0.71 *** (0.63, 0.80)	0.78 *** (0.68, 0.89)
Likely	Ref	-	-
**COVID-19 diagnosis status**			
No	Ref	-	-
Yes		1.17 (0.94, 1.45)	1.24 (0.98, 1.57)
**Household member COVID-19 positive diagnosis status**			
No	Ref	-	-
Yes		0.84 * (0.71, 0.98)	0.90 (0.76, 1.07)
**Social-distancing stress**			
Not at all stressful	Ref	-	-
Stressful		1.21 ** (1.06, 1.37)	1.16 * (1.01, 1.34)
**Anxiety/depression**			
No	Ref	-	-
Yes		0.67 *** (0.59, 0.76)	0.90 (0.78, 1.04)
**PTSD-5 qualifying status**			
No	Ref	-	-
Yes		0.91 (0.80, 1.03)	0.94 (0.82, 1.08)

Model I = Concerns only the sociodemographic characteristics. Model II = Excludes the sociodemographic characteristics. Model III = Combines Models I and II. The models were based on complete case analysis. AOR = Adjusted odds ratio. CI = Confidence interval. Ref = Reference GAD-2 = Generalized Anxiety Disorder. PHQ-2 = Patient Health Questionnaire-2. PHQ-4 = Patient Health Questionnaire-4. PTSD = Post-traumatic Stress Disorder. * Statistical significance level = *p* < 0.05. ** Statistical significance level = *p* < 0.01. *** Statistical significance level = *p* < 0.001.

## Data Availability

The data are not publicly available because the survey was collected for use by the Immigrant Health and Health Disparities (IHD) Research Laboratory, Division of Intramural Research, National Institute on Minority Health and Health Disparities at the National Institutes of Health. We have only recently started analyzing the data and cannot make them publicly available.

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
