# Peer review of "Differences in the Perceived Likelihood of Receiving COVID-19 Vaccine"

_ijerph, 2022, doi:10.3390/ijerph192113723_

Round 1

Reviewer 1 Report

This manuscript explored the perceived likelihood of persons receiving a COVID-19 vaccine. It was found that some segments of the population were more likely than others to express vaccination hesitancy. The results were not surprising and are consistent with other studies. This information provides important insights about where to target educational efforts. Understanding vaccination hesitancy is a critical issue; the manuscript is well-written and the research methods are appropriate. Before publication, I would recommend that the literature on vaccine hesitancy be updated. Some articles that could be cited include:

Peretti-Watel et al. - Lancet Infectious Disease (2020)

Sun et al. - Journal of Rural Health (2021)

Albrecht - BMC Public Health (2022)

Author Response

This manuscript explored the perceived likelihood of persons receiving a COVID-19 vaccine. It was found that some segments of the population were more likely than others to express vaccination hesitancy. The results were not surprising and are consistent with other studies. This information provides important insights about where to target educational efforts. Understanding vaccination hesitancy is a critical issue; the manuscript is well-written and the research methods are appropriate. Before publication, I would recommend that the literature on vaccine hesitancy be updated. Some articles that could be cited include:

Peretti-Watel et al. - Lancet Infectious Disease (2020)

Sun et al. - Journal of Rural Health (2021)

Albrecht - BMC Public Health (2022)

Response

Thank you for your feedback. We have worked to incorporate these articles into our article.

Reviewer 2 Report

It is an interesting study with a big population. However, I would have some suggestions I believe are worth considering:

Abstract:

In the abstract I would write clearly what the aim of the study is. I can assume that the intervention provided is the aim, but in order to evaluate the conclusions I believe it is better the state a clear aim of the study/objectives.

Adjusted Odds Ratio 20 (AOR)= 0.29-59 - it is not clear why the authors have provided a OR range, please provide the OR itself or please explain why did you decide to write the range

Introduction:

“To our knowledge, this is the first study to examine the association between per- 65 ceived likelihood of receiving the COVID-19 vaccine and COVID-19 diagnosis status, per- 66 ceived likelihood of SARS-CoV-2 infection, and diagnosis of anxiety/depression. Improv- 67 ing our understanding of these associations can aid in identifying priority population and 68 developing effective community-based strategies to improve vaccine education.” - I would rephrase this paragraph with a bigger emphasis on the objectives without referring to whether this is the first study in the field.

 I am wondering if the COVID-19 vaccine is available for free in the US, I think it would be beneficial for the background to add that information.

Methods:

If the survey was made only in English how can the authors be certain that the non-English speaking participants understood all the questions?

Discussion

The discussion is very interesting and well written.

Author Response

It is an interesting study with a big population. However, I would have some suggestions I believe are worth considering:  

Abstract:

  1. In the abstract I would write clearly what the aim of the study is. I can assume that the intervention provided is the aim, but in order to evaluate the conclusions I believe it is better the state a clear aim of the study/objectives.

Response

Thank you for the comment. We did not provide intervention as an aim of our study. The aims of our study now read: “There are limited studies on perceived likelihood of receiving COVID-19 vaccine among the general US population and its subpopulations. We examined the association between the perceived likelihood of receiving COVID-19 vaccine with the likelihood of contracting COVID-19, social distancing stress, COVID-19 diagnosis status, mental health disorders, and sociodemographic characteristics.”

Comment

  1. Adjusted Odds Ratio 20 (AOR)= 0.29-59 - it is not clear why the authors have provided a OR range, please provide the OR itself or please explain why you decided to write the range.

Response

We appreciate your comment. The AOR range was provided for the combined increasing age groups of 18-49 (i.e., 18-25, 26-34, 35-49) because the AORs reflect that pattern.

Introduction:

Comment

  1. “To our knowledge, this is the first study to examine the association between perceived likelihood of receiving the COVID-19 vaccine and COVID-19 diagnosis status, perceived likelihood of SARS-CoV-2 infection, and diagnosis of anxiety/depression. Improving our understanding of these associations can aid in identifying priority population and developing effective community-based strategies to improve vaccine education.” - I would rephrase this paragraph with a bigger emphasis on the objectives without referring to whether this is the first study in the field.

Response

Thank you for this observation. We have revised the beginning sentence as you recommended.

Comment

  1. I am wondering if the COVID-19 vaccine is available for free in the US, I think it would be beneficial for the background to add that information.

Response

Thank you. Yes, the COVID-19 vaccine is available for free in the US. We have incorporated your suggestion in the Introduction section.

Methods:

Comment

  1. If the survey was made only in English how can the authors be certain that the non-English speaking participants understood all the questions?

Response

Thank you for pointing out this important limitation. We mentioned this limitation in our limitation section.

Discussion

Comment

  1. The discussion is very interesting and well written.

Response

Thank you for the feedback.

Reviewer 3 Report

ABSTRACT

Acceptable

INTRODUCTION

Give an introduction regarding the situation in the US, whether the vaccine is free of cost, available, accessible, and how many compulsory doses were indicated, etc., for better readers’ understanding.

Line 58 “COVID-19 vaccination, however, may reduce depression/anxiety symptoms, particularly among the most vulnerable…” In this case, depression and anxiety are the outcomes. On the contrary, in the current study, the perceived likelihood of receiving COVID-19 is the outcome variable, and depression/anxiety are the independent outcomes. A clear conceptual framework should be available to determine which factors must be explored in determining/influencing the perceived likelihood of receiving COVID-19.

The objective, if it was as stated in lines 65-67, did not wholly reflect the areas covered in the study. This is obviously seen in the results section.

METHOD

Acceptable

RESULT

What might “Other” sexual orientation include?

Why do you need “Anxiety/depression” variable when you already have “Depression symptoms” and “Anxiety symptoms?”  This might introduce multicollinearity. Suggest removing “Anxiety/depression” variable.

Please do not repeat all the information in the tables in the text (page 6). Please summarize and remove the repetitions.

DISCUSSION

Line 223. Do not say that this is the first study when you have other articles reporting the same.

Line 242-243. What does “sexual minority individuals tend to experience stigmatization” have to do with perceived likelihood of receiving COVID-19 when it has nothing to do with sexual orientation? This again reflects the justification for selecting variables for this research, as indicated above.

Author Response

ABSTRACT

Acceptable

INTRODUCTION

Comment

  1. Give an introduction regarding the situation in the US, whether the vaccine is free of cost, available, accessible, and how many compulsory doses were indicated, etc., for better readers’ understanding.

Response

Thank you for the recommendation. We have incorporated this information in the introduction.

Comment

  1. Line 58 “COVID-19 vaccination, however, may reduce depression/anxiety symptoms, particularly among the most vulnerable…” In this case, depression and anxiety are the outcomes. On the contrary, in the current study, the perceived likelihood of receiving COVID-19 is the outcome variable, and depression/anxiety are the independent outcomes. A clear conceptual framework should be available to determine which factors must be explored in determining/influencing the perceived likelihood of receiving COVID-19.

Response

Thank you for the comment. The sentences were providing information on how anxiety/depression influences COVID-19 vaccination or hesitancy. As we pointed out, anxiety/depression increases the risk of COVID-19 hesitancy while COVID-19 vaccination reduces the impact of anxiety/depression. Hence, understanding the association between the perceived likelihood of receiving COVID-19 vaccine and anxiety/depression in the general population is important for addressing this risk factor and improving the COVID-19 vaccination. We would like to respectfully maintain this sentence for the current focus of this paper.

Comment

  1. The objective, if it was as stated in lines 65-67, did not wholly reflect the areas covered in the study. This is obviously seen in the results section.

Response

Thank you for the comment. We have revised the sentence to make our aim broader to reflect our results.

METHOD

Acceptable

Response

Thank you.

RESULT

Comment

  1. What might “Other” sexual orientation include?

Response

Thank you. We have changed the label of this category to “Something else” to avoid othering. In our study, “Something else” includes but is not limited to pansexual, asexual, demisexual, and other individuals who do not identify as heterosexual, lesbian, gay, or bisexual.

Comment

  1. Why do you need “Anxiety/depression” variable when you already have “Depression symptoms” and “Anxiety symptoms?” This might introduce multicollinearity. Suggest removing “Anxiety/depression” variable.

Response

Thank you for the comment. We agreed that could introduce multicollinearity. In Table 1, we presented all variables descriptive and bivariate distribution, but  in Table 2 or result sections, we included only “Anxiety/depression” variable in the logistic regression models to avoid multicollinearity.

Comment

  1. Please do not repeat all the information in the tables in the text (page 6). Please summarize and remove the repetitions.

Response

Thank you for this comment. We believe we did not repeat the results. We focused less on our descriptive and bivariate analysis results. We examined three models and presented only the significant results. The nature of the results reflected the models we examined. Hence, we would like to retain this information respectfully.

DISCUSSION

Comment

  1. Line 223. Do not say that this is the first study when you have other articles reporting the same.

Response

We appreciate your comment. We have revised the sentence.

Comment

  1. Line 242-243. What does “sexual minority individuals tend to experience stigmatization” have to do with perceived likelihood of receiving COVID-19 when it has nothing to do with sexual orientation? This again reflects the justification for selecting variables for this research, as indicated above.

Response

Thank you for the comment. That sentence was to emphasize possible reasons why sexual minority individuals had a high perceived likelihood of receiving the vaccine but may not in actuality receive the actual vaccine. Sexual minority individuals experience greater health barriers than heterosexual individuals due to stigmatization, abuse, harassment, and historical heteronormative biases. These barriers may significantly hinder vaccine uptake. 

Reviewer 4 Report

Dear Authors,

Thank you for the extensive work done on investigating the differences in the perceived likelihood of receiving COVID-19 vaccine amongst the adult population in the US. I have several questions and comments for your consideration:

1.       Abstract should be unstructured, without headings, about 200 words (as per instruction to authors by this journal). Also, the first sentence in the current abstract (lines 12-13) does not fall under objective.

2.       Lines 44-46: “Presently, over 25%...” suggest to put a date to this for better clarity (e.g. as of date xx…)

3.       Lines 50-52: Racial and ethnic disparities by Williams et al – suggest to add a brief highlight on the findings – which race/ethnic was higher/lower in the vaccination coverage?

4.       The objective mentioned in the abstract is slightly different from that specified in the introduction (lines 65-67). For one of the independent variable, “perceived” likelihood of SARS-CoV-2 infection, the “perceived” is important and should not be dropped in further mention/discussion of this variable.

5.       Methods – Was the data used for this study collected for another purpose? Was the present study part of another bigger/umbrella protocol or study?

6.       Were samples recruited from all over the US? How was the sample size derived from/at?

7.       An important question – for participants who answered the survey – have they received any COVID-19 vaccine prior to answering the survey? If they have received the COVID-19 vaccine prior to answering the survey, would the “perceived likelihood of receiving the COVID-19 vaccine” the appropriate measure to be investigated here?

8.       Statistical analysis – why the usage of 3 models for analysis?

9.       The models were based on complete case analysis – in Table 2, it would be good to include the final number of cases used for analysis in each of the models.

10.   “Missing” as a group for analysis in age groups in all models (Table 2) – how do you interpret this? Why were there, comparatively, so high proportions of missing for age groups compared to other variables?

11.   Results section write-up – it may be better to highlight some of the findings instead of describing again, in text, all the stats (percentages, AORs, CI) for every variable, since these stats are available in the Tables provided. The current write-up is very long and repetitive (of the stats given in the Tables).

12.   Lines 236-239: If the difference is not statistically significant, they are comparable.

13.   It is contradictory in some parts when authors suggest that nationally representative samples were used (e.g. in Lines 282-283) but in the limitation, it was stated that samples were not representative of the US. Who can the results be generalizable to then? Please do comment on the representativeness and generalizability of the findings from this study.

14.   Lines 317-323 fits more into future research directions rather than overall conclusion of the study.

Author Response

Thank you for the extensive work done on investigating the differences in the perceived likelihood of receiving COVID-19 vaccine amongst the adult population in the US. I have several questions and comments for your consideration:

Comment

  1. Abstract should be unstructured, without headings, about 200 words (as per instruction to authors by this journal). Also, the first sentence in the current abstract (lines 12-13) does not fall under objective.

Response

The abstract has been revised accordingly.

Comment

  1. Lines 44-46: “Presently, over 25%...” suggest to put a date to this for better clarity (e.g. as of date xx…)

Response

Thank you for your feedback. We have included a date (July 2022) to make the statement more clear.

Comment

  1. Lines 50-52: Racial and ethnic disparities by Williams et al – suggest to add a brief highlight on the findings – which race/ethnic was higher/lower in the vaccination coverage?

Response

This sentence has been revised accordingly.

Comment

  1. The objective mentioned in the abstract is slightly different from that specified in the introduction (lines 65-67). For one of the independent variable, “perceived” likelihood of SARS-CoV-2 infection, the “perceived” is important and should not be dropped in further mention/discussion of this variable.

Response

Thank you. We have revised the information to have one objective throughout the study. We dropped “SARS-CoV-2” and used COVID-19 to be consistent throughout the manuscript.

Comment

  1. Methods – Was the data used for this study collected for another purpose? Was the present study part of another bigger/umbrella protocol or study?

Response

The data used for this analysis was part of our bigger study. We’ve now made this clearer in our methods section. “Our study utilizes data from a larger cross-sectional survey that assessed general health status; COVID-19 symptoms, testing and prevention; chronic illness management; mental health; the pandemic’s economic impact; and participants’ sociodemographic characteristics. It was an online survey conducted among a national sample of US adults aged 18 years or older. This survey sought to (1) examine the effects of social and physical distancing on physical and psychosocial health and disparities among US adults (i.e., US-born & foreign-born); and (2) identify factors that potentially influence/explain the disparities”

Comment

  1. Were samples recruited from all over the US? How was the sample size derived from/at?

Response

Yes, the sample was collected in the general US population. We’ve detailed this in the method section: “The National Institutes of Health contracted Qualtrics LLC to conduct the recruitment and distribution of the online survey to the participants. Qualtrics used proprietary consumer panels to randomly sample participants with matching demographic characteristics to complete the survey. Oversampling was applied for low income (<$25,000 annual household income) and rural (self-reported residing in rural area, cross-reference with zip codes already collected by Qualtrics) adults among non-Hispanic White, non-Hispanic Black, Hispanic adults, and foreign-born participants.”

Comment

  1. An important question – for participants who answered the survey – have they received any COVID-19 vaccine prior to answering the survey? If they have received the COVID-19 vaccine prior to answering the survey, would the “perceived likelihood of receiving the COVID-19 vaccine” the appropriate measure to be investigated here?

Response

Thank you for this comment. The survey was conducted in the general population regardless of their COVID-19 vaccination status. The nature of the measure is appropriate to assess “perceived likelihood of receiving the COVID-19 vaccine” irrespective of the COVID-19 vaccination status.

Comment

  1. Statistical analysis – why the usage of 3 models for analysis?

Response

We appreciate your question. We used three models because we had main independent variables and covariates, so we wanted to assess their impact on the outcome variable under different conditions (adjusting vs not adjusting).

Comment

  1. The models were based on complete case analysis – in Table 2, it would be good to include the final number of cases used for analysis in each of the models.

Response

Thank you for this observation. We have now included the number of final cases for each model.

Comment

  1. “Missing” as a group for analysis in age groups in all models (Table 2) – how do you interpret this? Why were there, comparatively, so high proportions of missing for age groups compared to other variables?

Response

Thank you for the question. Currently, there were some irregular age values (e.g., “1”, “2”, “10”, etc.) that we could not ascertain so we had to set them to missing to ensure appropriate research conduct.

Comment

  1. Results section write-up – it may be better to highlight some of the findings instead of describing again, in text, all the stats (percentages, AORs, CI) for every variable, since these stats are available in the Tables provided. The current write-up is very long and repetitive (of the stats given in the Tables).

Response

We appreciate your comment and have cleaned up the results section accordingly.

Comment

  1. Lines 236-239: If the difference is not statistically significant, they are comparable.

Response

Thank you for your important point. We have amended the sentence to now read, “We also found gay respondents were more likely to report vaccination intent than heterosexual individuals. However, it should be noted their high perceived likelihood of receiving the vaccine may not be synonymous with vaccine access or eventual vaccination given sexual minority communities experience greater health barriers than heterosexual individuals [26-28]. Specifically, sexual minority individuals experience higher rates of stigmatization, medical abuse, harassment, and medical mistrust due to historical medical mistreatment, health services or information often not being tailored to their needs, and the lack of safe, affirming clinical and public spaces [27, 28]. Further studies, particularly longitudinal studies, with larger samples of sexual minority individuals should examine these issues further.”

Comment

  1. It is contradictory in some parts when authors suggest that nationally representative samples were used (e.g. in Lines 282-283) but in the limitation, it was stated that samples were not representative of the US. Who can the results be generalizable to then? Please do comment on the representativeness and generalizability of the findings from this study.

Response

We appreciate your comment. We have tried to make it clearer that we used a national sample, as opposed to a nationally representative sample. We could not use “nationally representative samples” statement because we did not apply sampling or survey weights in this study. Despite this, our study’s findings still have important implications on populations potentially at risk for lower vaccination rates.

Comment

  1. Lines 317-323 fits more into future research directions rather than overall conclusion of the study.

Response

Thank for the comment. We have moved these sentences into the discussion section.

Round 2

Reviewer 3 Report

Comment: Line 242-243. What does “sexual minority individuals tend to experience stigmatization” have to do with perceived likelihood of receiving COVID-19 when it has nothing to do with sexual orientation? This again reflects the justification for selecting variables for this research

The answer to this comment is not satisfactory. Please remove the statement.

Author Response

Comment

Line 242-243. What does “sexual minority individuals tend to experience stigmatization” have to do with perceived likelihood of receiving COVID-19 when it has nothing to do with sexual orientation? This again reflects the justification for selecting variables for this research. The answer to this comment is not satisfactory. Please remove the statement.

Response

We appreciate your continued follow-up regarding this part of our manuscript. To further clarify our reasoning, sexual orientation was included as a variable because sexual minority individuals are a uniquely vulnerable and at-risk population for not only SARS-CoV-2 infection, but also barriers to COVID-19 vaccine access and healthcare at large. It is true, sexual orientation does not inherently dictate likelihood of receiving the vaccine (i.e., just because someone is gay, it doesn’t mean they will automatically be less likely to vaccinate). However, the social and structural factors that uniquely hinder healthcare access (e.g., heteronormative bias, lack of LGBTQ-affirming spaces) for sexual minority groups can and do affect vaccination likelihood. For example, Azucar et al. found vaccine hesitancy among LGBTQIA adults in Los Angeles, California was attributed to historical and ongoing trauma, stigma, discrimination, and fear of violence both inside and outside clinical spaces (https://ajph.aphapublications.org/doi/full/10.2105/AJPH.2021.306599). Differences in COVID-19 prevention behaviors have also been reported between sexual minority and heterosexual individuals (https://www.liebertpub.com/doi/10.1089/lgbt.2021.0002). Finally, sexual minority individuals have higher rates of chronic disease (e.g. asthma, cancer, diabetes, heart disease, HIV, kidney disease), substance use, poor sleep, and mental health conditions—which are linked to minority stressors (e.g., discrimination, stigma) and increase the risk of COVID-19 morbidity, severe symptoms and complications, and mortality (https://www.cdc.gov/mmwr/volumes/70/wr/mm7005a1.htm and https://pubmed.ncbi.nlm.nih.gov/33887160/). Ultimately, understanding COVID-19 vaccine intention among sexual minority populations is essential given the limited information known on vaccine coverage, vulnerability to COVID-19 of this population, and significant barriers to healthcare access among sexual minority groups. We have included the following justification in our methods section: “Sexual orientation was included as a variable given the historical medical mistreatment and abuse faced by sexual minority groups that have been shown to affect COVID-19 vaccine uptake (Azucar et al., 2022)” (Page 3).

Reviewer 4 Report

Dear authors, 

Thank you for the substantial revision done. The current manuscript reads better to me. I have still some minor comments for your consideration. 

1. The abstract needs to be proof read. Certain sentences (lines 16-17) are hanging or not written in sentence form. 

2. Not sure if COVID-19 infection is the term you would want to use and be consistent in. COVID itself refers to a disease, I would think that SARS-CoV-2 infection is more appropriate but I leave it to your consideration.

3. My earlier comment #7 regarding vaccination status was clarified, but not reflected anywhere in the revised manuscript. Suggest to consider adding that clarification in.

4. Missing age, while clarified, is still quite substantial. Not sure if you would want to exclude the cases with missing age, like what was done to cases with missing information for other variables? Otherwise, I would suggest to include this as one of the limitations of the study. 

Otherwise, I have no further comments. Well done!

Author Response

Comment

The abstract needs to be proofread. Certain sentences (lines 16-17) are hanging or not written in sentence form.

Response

Thank you for the feedback. We have edited the abstract to fix these errors.

Comment

Not sure if COVID-19 infection is the term you would want to use and be consistent in. COVID itself refers to a disease, I would think that SARS-CoV-2 infection is more appropriate but I leave it to your consideration.

Response

We agree this should be changed. We have revised the manuscript accordingly.

Comment

My earlier comment #7 regarding vaccination status was clarified, but not reflected anywhere in the revised manuscript. Suggest considering adding that clarification in.

Response

Thank you for your feedback. We have included the following line in the limitations section: “Also, the survey was conducted in the general population regardless of their COVID-19 vaccination status. As such, individuals who have already received the vaccine may also be captured.” (Page 10).

Comment

Missing age, while clarified, is still quite substantial. Not sure if you would want to exclude the cases with missing age, like what was done to cases with missing information for other variables? Otherwise, I would suggest including this as one of the limitations of the study.

Response

Thank you for the feedback. We have included the missing information as a limitation in the limitation section. (Page 10)